# Use of Brain-Derived Stem/Progenitor Cells and Derived Extracellular Vesicles to Repair Damaged Neural Tissues: Lessons Learned from Connective Tissue Repair Regarding Variables Limiting Progress and Approaches to Overcome Limitations

**DOI:** 10.3390/ijms24043370

**Published:** 2023-02-08

**Authors:** David A. Hart

**Affiliations:** Department of Surgery and Faculty of Kinesiology, University of Calgary, Calgary, AB T2N 4N1, Canada; hartd@ucalgary.ca

**Keywords:** stem cells, pluripotent cells, neural stem cells, brain-derived progenitor cells, tissue regeneration, aging and stem cells

## Abstract

Pluripotent neural stem or progenitor cells (NSC/NPC) have been reported in the brains of adult preclinical models for decades, as have mesenchymal stem/stromal cells (MSC) been reported in a variety of tissues from adults. Based on their in vitro capabilities, these cell types have been used extensively in attempts to repair/regenerate brain and connective tissues, respectively. In addition, MSC have also been used in attempts to repair compromised brain centres. However, success in treating chronic neural degenerative conditions such as Alzheimer’s disease, Parkinson’s disease, and others with NSC/NPC has been limited, as have the use of MSC in the treatment of chronic osteoarthritis, a condition affecting millions of individuals. However, connective tissues are likely less complex than neural tissues regarding cell organization and regulatory integration, but some insights have been gleaned from the studies regarding connective tissue healing with MSC that may inform studies attempting to initiate repair and regeneration of neural tissues compromised acutely or chronically by trauma or disease. This review will discuss the similarities and differences in the applications of NSC/NPC and MSC, where some lessons have been learned, and potential approaches that could be used going forward to enhance progress in the application of cellular therapy to facilitate repair and regeneration of complex structures in the brain. In particular, variables that may need to be controlled to enhance success are discussed, as are different approaches such as the use of extracellular vesicles from stem/progenitor cells that could be used to stimulate endogenous cells to repair the tissues rather than consider cell replacement as the primary option. Caveats to all these efforts relate to whether cellular repair initiatives will have long-term success if the initiators for neural diseases are not controlled, and whether such cellular initiatives will have long-term success in a subset of patients if the neural diseases are heterogeneous and have multiple etiologies.

## 1. Purpose of the Review

Since the reporting of both Neural Stem Cells (NSC) and Mesenchymal Stem Cells (MSC) ~30 years ago, much research has been undertaken with them to both repair/regenerate brain tissues and connective tissues of the musculoskeletal system. As the brain and its regulation is likely much more complex than the corresponding connective tissues such as articular cartilage, tendons, ligaments, and skin, the latter research effort can provide some insights into similarities and differences with those targeting brain centres to impact chronic conditions such as Alzheimer’s disease, Parkinson’s disease, and multiple sclerosis, as well as more acute conditions such as stroke and brain trauma. Some conclusions that have arisen from the study of stem cell therapy, and their derivatives such as extracellular vesicles, to effect repair of damaged or diseased connective tissues may also have relevance to future research efforts focused on the brain. This review will attempt to discuss these similarities and differences, and then conclude with comments regarding areas for cross-learning in the two areas.

## 2. Introduction

Nearly 30 years ago in the 1990s, it was reported that cells with stem cell attributes could be isolated from the brains of adult rodents [1,2,3,4,5]. The cells were labelled NSC, and isolation of the cells and exposure to a “cocktail” of in vitro conditions containing growth factors led to the development of aggregates or organoids of these cells labeled “neurospheres” [6,7,8,9]. Such findings were confirmed and set off an extensive research effort to investigate how best to use these cells to potentially repair brain centres damaged by neurodegenerative diseases such as Parkinson’s Disease, Alzheimer’s Disease, and others such as Multiple Sclerosis, which affect so many people, as well as diseases which are projected to affect so many more individuals going forward [10,11] or brains affected by conditions such as Cerebral Palsy [12].

After several decades and millions of dollars in grant funding and foundation support, this effort to date has achieved rather limited success, much to the chagrin of millions of patients who had high expectations for a solution to their very debilitating conditions. Even at the local research microcosm, the initial findings led to the formation of a company (NeuroSpheres Ltd., Calgary, AB, Canada), but ultimately the effort was not successful, and the company failed. This microcosm of activity is somewhat representative of the overall effort to capitalize on the in vitro abilities of these neural stem/progenitor cells. While much of this effort was unsuccessful, it was valuable to help define what NSC were not capable of doing to repair and regenerate compromised brain structures, and while perhaps naïve in retrospect, filled a void in understanding what was needed to actually define their potential role(s) in brain development, maturation, and senescence.

Thus, much of this initial research effort to exploit the abilities of NSC to repair/regenerate neural tissues damaged by injury or disease was focused on using the cells without much concern regarding variables that could impact those abilities. Therefore, this approach perhaps succumbed to the hype of “stem cells” rather than an analytic approach following a thorough in-depth and thoughtful analysis of the factors that could impact the potential for success. However, in many respects, this experience with NSC paralleled that with MSC, a situation that led Caplan [13,14] to advocate for changing the name from mesenchymal stem cells to mesenchymal signaling cells. The >30 years of experience certainly have provided much information that should be considered going forward to both avoid and consider before attempting further exploration of the regenerative capacity of these cells. Likely consideration for variables to control or account for include those related to the cells, including age, sex-dependent epigenetic modifications, alterations induced by their culture conditions, and the cell-surface recognition systems used by the cells, as well as the environment the cells will be placed in to affect repair. These potentially also include which centres in the brain are to be targeted (i.e., do all centres of the brain have equal potential for repair and a return to functionality), presence of inflammation, number of cells of the affected tissues remaining and their functionality, co-morbidities, concurrent drug exposure, sex, and age. Given the unique interactions between centres in the brain, and the development of such interrelationships during early development and growth, the recovery of brain centres requires a return of both the structural integrity and the functional integrity via connections to other centres and peripheral locations due to the regulatory nature of some centres. Thus, the challenges to overcome current limitations will require addressing variables focused on both the cells and the environment. While some of these challenges are unique to the brain, others may also be similar to challenges to those of other stem/progenitor cells in other locations and tissues.

## 3. Parallels between NSC and Mesenchymal Stem Cells/Medicinal Signaling Cells in Outcomes and Challenges

The above discussion regarding NSC begs the question “are the findings with NSC unique to this population, or are they similar to experiences with other stem/progenitor cell populations to affect tissue repair and regeneration?” Based on the published literature, the effort with neural cells is likely not unique in that in some respects a similar path has been observed with MSC first reported by Caplan (reviewed in [15,16,17,18]). That is, the in vitro abilities of MSC from sources such as bone marrow, adipose tissue, or others to differentiate towards the chondrogenic, osteogenic, and adipogenic lineages by exposure to unique “cocktails” of growth factors, biochemical, media, and serum were seized upon by a myriad of researchers to exploit these abilities in vivo to effect repair and attempt to regenerate tissues such as cartilage, ligaments, tendons, and menisci that had been damaged by injury or disease [19,20]. However, analogous to the experience with neural “stem” cells, the in vivo experience with MSC was met with limited success after a significant investment and significant effort by researchers worldwide. This situation led Caplan [13,14] to suggest that MSC be renamed Medicinal Signaling Cells in response to the mounting evidence that in vivo MSC actually appeared to function primary via secretion of biologically active molecules and release of exosomes or extracellular vesicles containing a subset of cellular molecules that can exert regulatory effects on cells that take them up and disgorge their contents, including miRNA [21,22]. Such activities working in a paracrine manner would indicate that the main function of MSC in vivo may be as a regulatory cell working in unique biological and biomechanical environments [20,23].

Using MSC derived from synovial fluid (SF), Krawetz et al. [24] reported that when MSC from normal, but not osteoarthritic, knees were exposed to chondrogenic differentiating media, the cells aggregated in response to the stimulus. Further studies by Harris et al. [25] indicated this was due to the presence of the cytokine MCP-1 in the osteoarthritis knees. Thus, these MSC were influenced by the in vivo environment, and as such, inserting normal MSC into an abnormal environment may compromise their ability to exert their pluripotent abilities. Likely, a similar principle holds for the in vivo application of NSC, where implanting them into specific brain centres affected by disease would also compromise their ability to affect repair or regeneration. In both situations, it is likely that inflammatory processes are evident, and thus, the injected cells could be compromised.

While the company NeuroSpheres is no more (as are many others), the finding that NSC form aggregates (neurospheres) [6] is of interest as the aggregation likely indicates that the cells express some elements of a recognition system with a ligand and cognate receptor. As the NSC appear to express such a system without overt induction via their ability to aggregate, this is somewhat different than what has been observed with synovial fluid-derived MSC, which express a similar system after exposure to a chondrogenic differentiation medium [24]. While the recognition systems used in both instances is unknown, they may be of a more primitive lectin–cell surface glycoside system [26]. However, the ability of NSC to form aggregates in vitro leads to the question of why such aggregation does not spontaneously occur in vivo. This could be due to the presence of an inhibitor in vivo that blocks aggregation, or the in vitro culture conditions have resulted in the expression of a system that now permits aggregation.

Interestingly, rather than a static construct, these neurospheres comprised of NSC are dynamic, with the cells able to move within the construct and interact [6]. Whether the movement of the cells is directed as in a chemotactic response to a secreted molecule or random or stochastic remains to be determined. As NSC, similar to MSC [27,28], are very heterogeneous [29,30], it is possible that some NSC within a population do secrete such molecules. Whether MSC derived from synovial fluid can move within the aggregates that form after exposure to a chondrogenesis-inducing media [24] is thus far unknown.

One could address improved understanding of the systems involved and whether they are sugar-based by using a variety of free sugars in an attempt to complete with endogenous sugars for lectin sites. Talaei-Khozani et al. [31] reported there is lectin profile variation in MSC from different sources, and Dodla et al. [32] reported that lectin binding profiles among stem cells may serve as biomarkers for neural progenitor cells. In addition, Freund et al. [33] and Jin et al. [34] reported that gangliosides with different sugars can be used to identify subsets of MSC. Furthermore, surface glycans could also be addressed using cell surface proteomic approaches to better understand potential expression of lectins on the NSC or MSC, as well as other potential phenotypic biomarkers [35]. Additional support for a role for specific carbohydrates on the cell surface of brain cells comes from studies indicating that deficiencies in certain glycosyl transferases such as an alpha1,3-fucosyltransferase [36] can result in brain abnormalities in the cerebral cortex of mice. These authors suggested that this glycosyl transferase plays a role in the differentiation, migration, and maturation of neural precursor cells in the developing cortex.

A further question regarding the aggregation of NSC and differentiated synovial fluid MSC is whether they use the same or different recognition systems. There are a variety of lectins that could be expressed on cells, as well as a variety of simple and complex glycosides that could also be expressed. Using lectin phenotyping, Talaei-Khozani et al. [31] reported that MSC from different tissue sources express different lectin-binding phenotypes, so likely NSC and synovial fluid-derived MSC may also express different patterns of glycosides. While it is not yet known whether these glycoside differences play a functional role with the cells, one could determine whether these differences play a cell-specific role in aggregation. Thus, NSC and MSC from the same species could be add-mixed (with one source labeled in a manner to allow for tracing; [6]) and assessment of whether cells only self-aggregated or mixed aggregates could form can be made. Finally, there are some reports indicating that sugar recognition is important in the interaction of exosomes with MSC [37,38,39]. Therefore, sugar-based systems may be important in cell recognition and function for both MSC and NSC.

It is also of interest that aggregates of MSC can detach and move from one aggregate to another when cultured in bioreactors [40], and that NSC in neurospheres can move around in and within an aggregate [6]. Therefore, individual cells are not fixed within an aggregate. How they accomplish such movements is not known. Possible mechanisms include reversible expression of lectins on their surface, proteolysis of surface proteins, or inducible expression of glycosidases, if indeed the recognition system involves lectin–glycoside interactions. Whatever the mechanisms involved, clearly the interaction of both NSC and MSC in aggregates is dynamic, and the cells may share a common mechanistic recognition system but use different components of the system.

Another parallel between MSC and NSC is their relative effectiveness in addressing tissue repair is that NSC/NPC and MSC may be more effective in acute situations in both the brain [41,42,43,44,45,46] and in connective tissues such as articular cartilage [47,48] than in chronic disease situations involving either the brain or connective tissues [23,49,50,51]. The latter may relate to the in vivo environments being very different, including the involvement of a disease process involving chronic inflammation and tissue degradation fragments, and thus neither cell source may function properly in the face of a chronic inflammatory process. Interestingly, parallels between regeneration of neuro tissues and cardiovascular regeneration have also been raised [52], and inflammation may again play a role in the two situations. As Mitrecic et al. [52] discuss in detail, use of stem cells for tissue repair may be effective when used in acute situations such as ischemic events such as those involving the heart or the brain. However, the heart muscle is less complex than the brain centres affected by neurogenerative conditions, so the repair of the heart may be more analogous to other connective tissues than the brain. While NSC appear to have some distinctive antigens, how they relate to specific functions in the brain is not well defined. In addition, as discussed by Mitrecic et al. [52], both cardiomyocytes and neural cells can be derived from MSC by cell-specific differentiation protocols, and thus can share their cell of origin. Thus, sharing experiences regarding attempts to repair/regenerate tissues with different cells with stem-like properties may eliminate some redundancy in the research effort going forward and help to focus some of the research effort on commonalities and then tissue-specific aspects of the repair.

Another parallel between NSC and MSC is their apparent sex differences. Differences between the sexes in tissue repair have been reported (reviewed in [53]) and sex-specific aspects of stem cells may contribute to such differences [54,55]. This concept is further supported by findings that estrogen can influence stem cell behavior [56] and androgens can influence neural progenitors [57,58]. Thus, the use of both NSC and MSC exhibit potential sex differences and their effectiveness could likely be influenced by the stage of life they are isolated from and the in vivo environment they are transplanted into to initiate repair. This consideration of sex and age has not always been evident in studies, but a failure to acknowledge their influence could contribute to outcomes.

## 4. Are NSC Similar to MSC and Behave Similarly Because They Are Similar, or Is it Because of the Circumstances under Which They Are Being Cultured and Assessed?

From the above discussion, there are several similarities between NSC and MSC in their behaviour, mainly in vivo. This may be due to the cells actually being very similar, or due to the fact that researchers have made them appear similar, in part due to how they are cultured in vitro, but also due to the circumstances of the models used to assess their potential in vivo!

MSC and NSC may behave similarly because they are similar in origin and other aspects, but only differ as a population in some details. Relevant to this point is the finding that MSC isolated from different tissues (i.e., bone marrow, synovial fluid) exhibit a preference for different lineages when induced to differentiate in vitro [27]. That is, bone marrow-derived MSC preferentially differentiate towards the osteogenic lineage while those from synovial fluid preferentially differentiate towards the chondrogenic lineage [27,59] Thus, the differences between NSC and MSC may reside in the tissue of origin rather than any fundamental differences. In preclinical mouse models of Alzheimer’s disease, implantation of NSC or MSC were both beneficial, but some differences were noted [60]. Interestingly, both cell types exhibited anti-inflammatory effects.

In contrast, NSC and MSC could be different at multiple levels but only appear to be similar due to artifacts of preparation and the in vivo models they are being used to assess their potential. Thus, removal of either NSF or MSC from their in vivo environments and culturing them in vitro under very artificial conditions (2D cultures, artificial medium, abnormal oxygen tension, possible presence of serum or some mix of reagents found to empirically enhance growth of the cells, absence of other cell types that may offer support, etc.) could induce either dedifferentiation or alterations that obscure initial differences. Certainly, NSC and MSC are heterogeneous [27,61], but some of the heterogeneity of MSC in vitro has been postulated to be artefactual [26,28,62]. However, chronic culturing under artificial conditions could also lead to epigenetic alterations that may not be reversible, or even mutations. Thus, NSC and MSC could appear to be similar after culturing for several passages due to the in vitro culture conditions which may select for cells with the ability to propagate in these very artificial conditions.

NSC and MSC could also appear to be similar due to the environments they are placed into during in vivo studies. Two aspects of the in vivo environments could shape the ability of in vitro propagated cells to be successful after implantation in vivo. The first is an inflammatory or catabolic environment resulting from an injury or disease process (discussed in [20,23]). The second is that the NSC or MSC are delivered to a site that is deficient or devoid of a threshold of endogenous cells and thus there is no template remaining, nor endogenous cells that can be stimulated to reform or regenerate the tissue of interest. Thus, the cells are being used late in a disease or pathologic process rather than earlier when there may be residual endogenous cells remaining to allow for effective signaling.

Implanting or injecting MSC into sites of musculoskeletal (MSK) tissue injury or disease means inserting them into an inflammatory environment as an injury leads to an inflammatory response whose purpose is to facilitate endogenous healing via a fibrotic process or an acute response that can proceed to become chronic. Thus, the environment is initially both catabolic and inflammatory, but subsequently should become more anabolic unless the process becomes chronic. Isolation of human MSC from the SF of an osteoarthritic knee leads to MSC with altered characteristics [24], and exposure of normal MSC derived from normal SF to SF from an osteoarthritic knee leads to the same alterations [25], presumably via MCP-1 [25]. Similarly, in an ovine model, Ando et al. [59] reported that synovial fluid MSC from an injured knee were compromised and it was likely due to the inflammatory mediator interleukin-1 (IL-1). Thus, an inflammatory process is occurring in the OA knee and this environment can alter the characteristics of normal MSC, possibly derived from other tissue compartments for autologous treatment or from allogeneic sources and put into the knee, and potentially altering their ability to repair/regenerate the tissues damaged by this chronic and progressive condition.

Additional studies have indicated that treatment of a knee with glucocorticoids immediately after a surgery to a knee can prevent the development of inflammation and subsequent osteoarthritis-like joint changes in preclinical models such as rabbits [63,64] and pigs [65]. Thus, surgery and/or an injury to a joint can lead to inflammation that can induce OA-like changes to the tissues and become persistent. Therefore, it may be necessary to inhibit an ongoing inflammatory response before considering implanting stem cell preparations or treating an ongoing inflammatory environment with an appropriate intervention to block the catabolic influence of the environment on the ability of the implanted cells to function properly. The latter may be more relevant to the real-life situation, but it may be less effective as a chronic inflammatory state may induce changes in both the injured tissue and associated uninjured tissues in a joint. Therefore, changing a catabolic environment to be more anabolic could also enhance the landscape to foster a better outcome and improved realization of the potential of the stem cells that are implanted. In some circumstances, implanting tissue-engineered constructs (TEC) into human [47,48] or porcine [66,67] chondral defects led to good outcomes with effective repair of the tissue. In these studies, there were no attempts to control endogenous inflammation or that arising after the surgery. However, while good outcomes were obtained, the regeneration process was not perfect [68,69], so perhaps further improvements could be obtained through judicious use of drugs or other interventions to control catabolic influences, or improved understanding of any growth and maturation-associated events that might be difficult to replicate in an adult environment. However, in this case, repair/regeneration does not recapitulate development.

Therefore, in neurodegeneration conditions accompanied by cell death via necrosis, release of pro-inflammatory molecules or degradation fragments, or induction and release of pro-inflammatory mediators, may also create an environment that is not conducive for optimal implantation of neuroprogenitor cells or neuro-organoids to initiate repair and regeneration of damage neuro-tissues. Without attenuating, such a catabolic environment very likely would diminish the chance for successful repair of the compromised brain tissue. Interestingly, in recent preclinical studies, de Munter et al. [70] reported that bone marrow-derived stem/stromal cells exerted more of an anti-inflammatory effect on neurodegeneration conditions than did anti-inflammatory drugs. Thus, mesenchymal stem cells could have an immunomodulatory role to play in diseases of the brain [71]. Furthermore, Nebie et al. [72] also recently discussed the potential for extracellular vesicles from platelets to improve neurological disorders. Thus, perhaps merely disrupting the catabolic inflammatory environment associated with a neurodegenerative disease could inhibit progression and further loss of cells, and if intervention was initiated early, retain sufficient endogenous cells to allow for complementary interventions to permit the endogenous cells to potentially repair themselves. Whether such facilitation of endogenous repair would exhibit age-related challenges to success should also be a focus of future research.

## 5. Is There an Influence of Age on the Success of NSC Implantation?

Many, if not most, degenerative diseases are diseases of aging irrespective of whether bone and joint, cardiovascular or neural. As in most situations patients prefer to use their own autologous stem cells, the cells that are often isolated and expanded in vitro are therefore from older individuals as well. Thus, “old” cells are being re-introduced into older patients and the question arises as to whether this scenario is contributing to the lack of success in some clinical trials, as stem cell numbers and functionality appear to decline with age [73,74,75]. Interestingly, many preclinical models use participants that are not aged, and thus there may be a disconnect between the translation from preclinical models to real life patient populations.

Recently, it was suggested that perhaps a primary role for MSC was in the young rather than the elderly [19], and that MSC from older individuals were compromised due to epigenetic alterations induced during life experiences and other aging contributions. This perspective is supported by considerable literature on the subject [19], and thus, it is also likely that a similar construct holds for NSC as well as MSC (assuming they are different in some respects). Therefore, unless it is possible to “refresh” stem cells from older individuals, the construct of using autologous cells from elderly patients may not lead to the successful outcomes that are needed. The alternative is to either use autologous cells taken when young (i.e., cord blood or Wharton’s jelly cells) and then stored frozen until needed, or using standardized allogeneic cells from young donors. While these latter approaches may be ethically acceptable for MSC, if NSC are specifically needed due to compartment characteristics, this may pose an ethical problem. If approaches to optimize NSC potential do not come to fruition, it may be necessary to use appropriate induced pluripotent stem cells or other approaches with neuro potential instead (discussed below).

## 6. Limitations Regarding Use of Brain-Associated NSC

While cells with the characteristics of “stem or progenitor” cells can be isolated from the adult brain of many species, their density and function declines with age [76,77], a time in life that many people would need them for regeneration of brain elements that are damaged by disease. Certainly, such cells may be “primed” to contribute to neural regeneration as being in a specific location can lead to adaptations (i.e., lectin phenotypes, epigenetic signatures; discussed in [26]) and thus may be destined to serve specific roles in the brain. Further to that point, given the complexity and diversity of the centres and structures in the brain and their integration, it is unclear presently as to whether there are “general” NSC/NPC [78], or there may be centre or structural-specific NSC/NPC that are associated with different elements within the brain. If the latter, then either the “general” versions can differentiate into such specific cell types in an environment-specific manner, or these “general” NSC/NPC serve a different function intrinsically from those associated with specific centres or structures.

Another limitation of their use is actually retrieving autologous NSC or NPC in sufficient number without damaging the brain. Even if small numbers could be obtained in some ethically approved manner, that would require extensive expansion in vitro under defined conditions to obtain quantities of cells that would potentially be required for clinical or preclinical applications. In addition, a number of reports [62,79,80,81,82] have indicated that expansion of MSC in vitro can contribute to the generation of cellular heterogeneity, and thus, the cells after expansion may no longer reflect the population that was initially isolated and their properties and abilities after implantation in vivo compromised. If only small numbers of MSC/NPC can be obtained, for them to be used in preclinical or clinical applications will require extensive expansion of the cells to achieve sufficient numbers, potentially generating heterogeneity that could compromise effectiveness to treat neurodegenerative conditions. Presently, there is no indication that NPC are different from mesenchymal stem cells in this regard. In addition, with mesenchymal stem cells in a preclinical model, it was shown that autologous cells taken from an inflamed joint were altered even after culturing [59], and thus, if MSC/NPC were to be used in an autologous manner from a patient with a neurodegenerative condition, the cells may also be altered and potentially compromised due to exposure to an inflammatory or catabolic environment associated with a chronic disease activity.

As an alternative to using NSC/NPC, it is possible to appropriately differentiated stromal/stem cells isolated from other environment alone (i.e., MSC) or in combination with NSC [83], induced pluripotent stem cells (iPSC) derived from an appropriate somatic cell [84,85,86,87], or extracellular vesicles (EV) derived from MSC or a relevant cell cultured under optimal in vitro culture conditions [88,89]. The latter are becoming an important avenue of research as they are not immunogenetic, can contain a range of molecules that can enhance endogenous cell repair of the damaged tissue, and could therefore be used in non-autologous application, which would overcome potential limitations associated with autologous materials.

While potential strengths and limitations associated with iPSC are evident [90,91,92,93], the use of EV would overcome or bypass many of those limitations. The potential of EV to facilitate repair in acute neurological conditions and in chronic conditions have been reviewed recently by a number of authors [94,95,96,97]. As their effectiveness depends in part on their content, the ability to influence their content by altering the in vitro conditions they are generated from also provides flexibility in their application [88,89]. Furthermore, the use of allogenic EV would also overcome concerns regarding EV derived from the brain of patients with neurodegenerative conditions [98].

## 7. Are the “Right” Models Being Used to Evaluate the Effectiveness of NSC?

As mentioned above, many preclinical models used relatively young animals that did not correspond with the ages of patients who needed such cells (MSC and NSC) to repair conditions such as dementia and other neuro diseases, as well as connective tissue diseases such as osteoarthritis. In addition to age, another aspect of preclinical clinical trials that may compromise outcomes is the use of “new” or experimental interventions late in the disease process. The medical model that is usually imposed is that one has to fail usual treatment before new and experimental approaches are initiated. Thus, if some of the role of NSC (and MSC) is that of a signaling cell via release of exosomes or extracellular vesicles containing critical factors which lead to recovery of remaining tissue-specific cells, then waiting too long to introduce them will likely compromise the chance for success when levels of such tissue-specific cells fall below a threshold required for recovery.

Relevant to the above point is also the nature of the induction of the tissue damage in model systems compared to those occurring “naturally” in patients. Thus, in patients, most of the time the conditions are chronic before diagnosis and the condition has achieved a threshold required to diagnose the condition. The inductive event could be autoimmune, a toxin, overt injury, or in the case of a subset of females, menopause (discussed in [99]). In most cases in patient populations, the inductive event is idiopathic or unknown. In contrast, preclinical models are usually not idiopathic, and the inductive event may be artificial and designed for a rapid induction which lends itself to assessment. An example of the latter is induction of a Parkinson-like disease due to the toxicity of MPTP (1-methyl-4-phenyl-1,2,3,6-tetrahydropyridine) for the appropriate cells in the rodent [100,101] or human brain [102]. Thus, this model is an acute insult, while the disease in patients is likely of a more chronic nature.

## 8. Do NSC Have a Primary Role as Regulatory Cells Rather than a Direct Regeneration/Cell Replacement Role?

Recently a different name [Pluripotent Mesenchymal Regulatory Cells, PMRC] has been proposed for what were called MSC to reflect the in vivo functioning to secrete molecules and release exosomes and the in vitro pluripotency to differentiate toward specific lineages [103]. Furthermore, it was suggested that the focus on the “stemness” of the cells may have been misinterpreted and the key feature is that the cells exhibit pluripotency. For reasons that still remain elusive, if the cells often do not exhibit their “stemness” in vivo, but still exhibit pluripotency in vitro, this pluripotency may reflect some additional functioning of the cells in conjunction with their signaling functions in vivo. Thus, while the in vitro “stemness” of these cells can still be exploited for tissue engineering purposes, it is somewhat of a “red herring” with regard to generating expectations for in vivo results.

The same argument may also hold for NSC. Similar to MSC, injection of free NSC into sites of neural damage has not led to highly reproducible regeneration of brain centres. While such findings could be interpreted as a failure to function in the damaged environment, it could also mean the cells cannot actually repair the damage directly. However, they could attempt to facilitate endogenous cells to enhance repair. If this option was their true role, then the ability to enhance repair would depend on the number and functionality of the remaining endogenous cells. If one waited until the damage has progressed beyond some threshold, there may not be sufficient cells to reverse the damage. Intervening early would be required rather than waiting until the local conditions were beyond such a threshold point. Unfortunately, most medical approaches involving new treatment opportunities usually wait until it is nearly end-stage disease, an approach that may likely compromise the use of NSC and MSC to elicit success in facilitating repair or regeneration.

## 9. The Way Forward

Given the parallels between PMRC/MSC and NSC, perhaps one should reconsider the name and instead use the term Pluripotent Neural Regulatory Cells [PNRC] and re-focus some of the research effort to better understand how the pluripotency exhibited in vitro is translated into in vivo functioning of these cells. As the individual centres in the brain likely each have unique environments associated with their integrated functioning, perhaps the pluripotency is related to the ability of these cells to adapt to individual environments to provide support for the health of cells in such centres or structures in a unique paracrine manner. Thus, PNRC could adapt via their pluripotent abilities to “differentiate” in subtle ways to release the unique combination of molecules delivered via secretion or release of exosome containing those specific molecules that are needed in that environment. Furthermore, the known immunomodulatory abilities of PMRC [70,103] may also be elaborated by NSC [104,105] to control minor endogenously generated or exogenously generated inflammatory stimuli that could pose risks for loss of integrity.

Thus, the change in thinking is that the role of PNRC is not to primarily replace cells that have been lost due to injury or disease, but to augment the survival and health of the residual cells in specific centres, under conditions that require a minimal or threshold number of residual cells to restore or partially restore function. There are at least two interesting points associated with this perspective; (1) there could be specific loss of PNRC numbers or functioning with aging [106] and life-span transitions, which would put the health and functioning of the other cells in a brain centre at risk; (2) it would be better to attempt to restore function earlier rather than later in a disease process when the number of residual cells is still prominent. Regarding Point 1, it is known that the numbers and function of PMRC/MSC decline with age [107,108,109] and the brain-associated NSC as defined by Reynolds et al. [1,2] act as a reservoir of pluripotent cells for replenishment of small numbers of NSC that are required in a subclinical manner for, perhaps, maintenance of tissue integrity. Regarding Point 2, the medical model of relying on drugs during the early stages of disease and only entertaining serious, potentially more risky interventions until late in the disease process actually works against the concept of restoration of function when there are sufficient residual cell numbers in the affected tissues to make the restoration of function with NSC more achievable. Secondly, there is the risk that with increasing time, the disease process becomes more chronic in nature, accompanied by epigenetic modulation or involvement of inappropriate cells, a situation that may preclude effective restoration of function by NSC. For example, the disease process for Parkinson’s Disease (PD) or Alzheimer’s Disease (AD) likely starts long before functional deficits can be detected. Regarding AD, the research focus has been strongly emphasizing removal of proteins that are believed to be responsible for the pathology and functional deficits (i.e., tau and tau fragments). One could debate whether this emphasis is on the causes of AD versus the outcome of the process, but the point is there is a time when recovery/restoration may become futile due to a lack of cells that can be engaged to benefit from the effects of PNRC! Thus, perhaps one should entertain the inversion of the treatment pyramid when it comes to considering the use of PNRC to interfere with disease processes such as AD or PD.

An additional variable in age-related dementia is that ~70% of patients are post-menopausal females (discussed in [99,110,111,112,113]). In ~50% of these cases, the condition appears to be vascular in nature, and it is not clear whether exposure to NPC would be directed to the vascular cells possibly affected by menopause (discussed in [99]), or the other cells. Interestingly, vascular cells can also produce tau tangles [99], one of the targets for therapy in AD [114,115]. Liu-Ambrose and colleagues [116,117,118,119] have discussed the ability of exercise in perhaps pre-menopausal females and beyond to prevent or inhibit the vascular-related changes that may lead to dementia or loss of cognition, implying that the cognitive changes in the brain cells may be indirect and associated with loss of vascular integrity in a subset of post-menopausal females. Sex differences in loss of cognition may therefore result from sex differences in regulation of the vascular component of specific tissues in the brain [120,121] during the aging process and vascular senescence [122,123]. As endothelial cells are heterogenous and appear to form a specific paracrine relationship with cells in different tissues [124,125,126], variation in the vascular contribution to different forms of dementia or cognition loss may occur. Thus, there may be a need to target unique vascular beds with MSC or NSC, or their EV to restore vascular integrity rather than just focus on the neural cells. The contribution of the vascular component to disease development and progression likely is more relevant to the brain and NSC than connective tissues where perhaps it is most relevant to bone but not as much to other tissues (discussed in [99]).

Finally, there is a caveat to considering the use of NSC to restore function in diseases such as AD or PD. That is, unless the causes of such diseases are better understood, the use of interventions such as NSC alone, or optimally produced EV with an appropriate cargo for repair of that particular target tissue, may end up being a “temporary” fix unless the fundamental initiators of disease are blocked. The approach may “buy” many patients time for the above to be accomplished, but it may not be a permanent fix. Related to the above point is one that may be more addressable, and that is the catabolic environment (potentially inflammatory) generated by the disease process. To better enhance potential for success in using NSC, such circumstances may require a coordinated negation of the catabolic environment in combination with the use of the NSC. This would not be unique to the brain as it should also be a consideration for the use of PMRC/MSC in inflammatory diseases of connective tissues [127].

## 10. Conclusions

There has been approximately 30 years of experience in the use of neural stem cells or neural progenitor cells to facilitate repair and regeneration of compromised brain tissues or MSC in the treatment of musculoskeletal tissues, and the success rate for impacting disease progression in major diseases has been limited. Thus, endogenous NSC/NPC and MSC cannot overcome disease development and progression and added larger numbers of cells cannot apparently overcome an altered environment due to the disease, or this is not their primary function. Given the complexity of the brain and its regulation, some conclusions arising from these studies can be put forward.

Both MSC and NSC may be more effective in addressing acute events leading to loss of tissue integrity in tissues of both systems.NSC and MSC may share many features, indicating that using NSC alone may not be sufficient to affect repair of compromised neural tissues.Factors such as inflammation associated with chronic degenerative diseases or conditions likely need to be addressed to enhance the efficacy of stem/progenitor cell therapies. Even though MSC and NSC have immunomodulatory abilities, there may be a need to adequately prepare the in vivo environment to induce an anabolic environment to facilitate the effectiveness of the implanted cells.Autologous NSC/NPC and MSC from aged patients or those with co-morbidities may themselves be compromised, and thus contribute to a lack of successful cellular interventions.Use of extracellular vesicles (EV) from appropriately cultured NSC/NPC or MSC/PMRC to ensure an optimized content would overcome some of the limitations of Points 3 and 4 in that they could be used as allogenic interventions.The use of EV requires a sufficient quantity of residual tissue cells to affect the regeneration/repair of the target tissue and influence a return to functionality. Therefore, interventions with EV would need to be initiated early in the disease process to allow for such regeneration, as delaying too long would both advance the inflammatory state and deplete the number of potential endogenous cells to affect repair.Due to the complexity of the brain compared to connective tissues, reconstituting the brain tissue also requires effective regulatory integration that may not be relevant to connective tissues.

Implementing some of the above-listed points may allow for the design of studies to address the many neurodegenerative diseases that affect so many individuals in ways that extend beyond those affected by diseases and conditions of the MSK system. However, there are caveats to the long-term success of this research effect. The first is whether there will be long-term success if the underlying disease mechanisms are not identified and addressed, as repair in the face of an on-going disease process may only be temporary. Secondly, diseases such as Alzheimer’s, Parkinson’s, and multiple sclerosis may have more than one etiology, and regenerative efforts may only offer long-term relief in a subset of patients. The analogy in the musculoskeletal field is osteoarthritis, which is an umbrella term for subsets of patients (reviewed in [128]), and thus this complexity was a contributing factor in the lack of success in developing single interventions to address the condition. Likely, diseases of the central neural system may also be heterogeneous in etiology, and this may complicate achieving success using cellular therapies.

## Data Availability

Not applicable.

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
