# Peer review of "Use of Brain-Derived Stem/Progenitor Cells and Derived Extracellular Vesicles to Repair Damaged Neural Tissues: Lessons Learned from Connective Tissue Repair Regarding Variables Limiting Progress and Approaches to Overcome Limitations"

_ijms, 2023, doi:10.3390/ijms24043370_

Round 1

Reviewer 1 Report

line 9 -correct be to been

NSC/NPC and MSC acronyms are not defined in the abstract, which is important as the abstract needs to stand alone

line 97 - change is to are

some of the information in lines 130-136 is duplicated in lines 147-152. It may be good to revise these two paragraphs, so that the information is only distributed once. The repeated information seems most pertinent to the latter paragraph, where specifics about a lectin-based aggregation mechanism are covered, and is less necessary for the first paragraph, which merely introduces the topic of aggregation.

could the author elaborate more on the commonalities between MSCs and NPCs as well as the "parallels between regeneration of neuro tissues and cardiovascular regeneration" discussed in lines 189-19? What specifically are the points of commonality that have been identified and how might they be capatilized on going forward to improve regenerative therapy approaches? The author raised a good point here and elaborating on it in this paragraph would enhance the impact of this section of the article.

The author raises an excellent point regarding the effect of in vitro culture conditions on the properties of MSCs and NSCs.

Some inconsistencies the italicization of in vitro and in vivo throughout that need correction.

The author raises several good points in terms of the therapeutic limitations and challenges to apply MSCs and NSCs. Overall, this is thoroughly reviewed and well written. The author appropriately cites relevant literature and provides a comprehensive overview that will benefit researchers working on these tools for regenerative therapies in developing collective thoughts and overcoming the challenges outlined by the author.

The conclusions are well put forward and succinct. This is a very satisfying read which raises many excellent points and provides thoughtful commentary on those points. Well done.

Reviewer 2 Report

This review discusses the similarities and differences between neural stem cells and mesenchymal stem cells, and how this knowledge can help in the application of cell therapy to regenerate brain structures. It highlights the heterogeneity of neural diseases and the complexity of the brain, and critically evaluates the translation of in vitro results to the in vivo situation.

The review is very well written and I have only a few very minor comments.

1/ Some abbreviations are not explained (MSK), some are mentioned more than once (NSC). 2/ I am also not quite sure what the abbreviation FCAHS means among the authors. Does it mean "Fellow of the Canadien Academy of Health Sciences"?

Reviewer 3 Report

This review paper summarized the similarities and differences in the applications of NSPC and MSC and discussed the lessons and potential approaches that could be useful for enhancing the progress in the application of cellular therapy to facilitate neural repair and regeneration. In general, this review paper is well-prepared and has a clear, logical structure.
